# Automating the Refinement of Reinforcement Learning Specifications

**Tanmay Ambadkar**[*]
The Pennsylvania State University
tsa5252@psu.edu

**Đorđe Žikelić**[†]
Singapore Management University
dzikelic@smu.edu.sg

**Abhinav Verma**[‡]
The Pennsylvania State University
verma@psu.edu

## Abstract

Logical specifications have been shown to help reinforcement learning algorithms in achieving complex tasks. However, when a task is under-specified, agents might fail to learn useful policies. In this work, we explore the possibility of improving coarse-grained logical specifications via an exploration-guided strategy. We propose AUTOSPEC, a framework that searches for a logical specification refinement whose satisfaction implies satisfaction of the original specification, but which provides additional guidance therefore making it easier for reinforcement learning algorithms to learn useful policies. AUTOSPEC is applicable to reinforcement learning tasks specified via the SpectRL specification logic. We exploit the compositional nature of specifications written in SpectRL, and design four refinement procedures that modify the abstract graph of the specification by either refining its existing edge specifications or by introducing new edge specifications. We prove that all four procedures maintain specification soundness, i.e. any trajectory satisfying the refined specification also satisfies the original. We then show how AUTOSPEC can be integrated with existing reinforcement learning algorithms for learning policies from logical specifications. Our experiments demonstrate that AUTOSPEC yields promising improvements in terms of the complexity of control tasks that can be solved, when refined logical specifications produced by AUTOSPEC are utilized. https://ambadkar.com/autospec

## 1 Introduction

Reinforcement Learning (RL) algorithms have made tremendous strides in recent years Sutton & Barto (2018); Silver et al. (2016); Mnih et al. (2015); Levine et al. (2016). However, most algorithms assume access to a scalar reward function that must be carefully engineered to make environments amenable to RL—a practice known as reward engineering Ibrahim et al. (2024). This creates challenges in applying RL to new environments where useful reward functions are hard to construct. Furthermore, scalar Markovian rewards cannot provide sufficient feedback for certain tasks Abel et al. (2021); Bowling et al. (2023), leading to growing interest in non-Markovian reward functions Li et al. (2017a); Jothimurugan et al. (2021); Alur et al. (2023).

To make non-Markovian rewards tractable, it is standard to represent them via logical specification formulas that capture the intended task. These approaches, known as specification-guided reinforcement learning Aksaray et al. (2016); Li et al. (2017b); Icarte et al. (2018); Jothimurugan et al. (2019; 2021), derive reward functions from logical specifications. However, this creates two challenges: (i) providing specifications granular enough to guide RL algorithms, and (ii) defining accurate labeling functions mapping environment states to specification predicates. Users often create

---

[*]Department of EECS, The Pennsylvania State University
[†]School of Computing and Information Systems, Singapore Management University
[‡]Department of EECS, The Pennsylvania State University

coarse specifications or labeling functions that, while logically correct, provide insufficient guidance for learning.

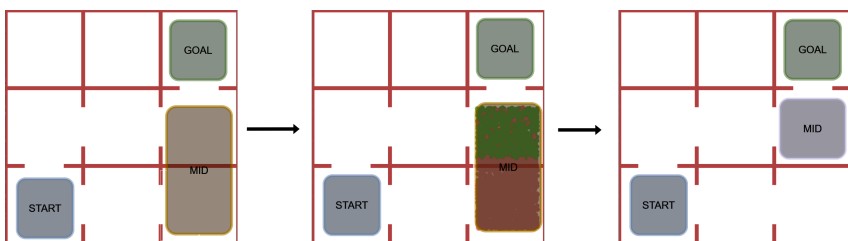

Figure 1: Example of refinement by AUTOSPEC in a 9-rooms environment. The original MID-node region includes a trap state from which recovery is impossible. The refined specification excludes this trap, enabling the agent to learn a policy with higher satisfaction probability.

We present AUTOSPEC, a framework for automatically refining coarse specifications without user intervention. We say that a logical specification is *coarse* (or *under-specified*) if its predicate labelings or logical structure are too coarse to allow specification-guided RL algorithms to translate logical specifications into reward functions that allow for effective learning of RL policies. AUTOSPEC starts with an initial logical specification, translates it to a reward function, and attempts to learn a policy. If the learned policy's performance is unsatisfactory, AUTOSPEC identifies which specification components cause learning failures and automatically refines both the specification formula and labeling function. The refined specification's satisfaction implies the original's satisfaction while providing additional structure for learning. This process repeats until a satisfactory policy is learned.

AUTOSPEC works with SpectRL specifications; boolean and sequential combinations of reach-avoid tasks Jothimurugan et al. (2019). Any SpectRL specification decomposes into an abstract graph where edges specify reach-avoid tasks Jothimurugan et al. (2021). AUTOSPEC identifies problematic edges and applies targeted refinements: either modifying the labeling function for regions (Figure 1) or restructuring the graph to add alternative paths. These problematic edges are identified by employing an *exploration-guided strategy* that utilizes empirical trajectory data to identify edges in the abstract graph whose reach-avoid tasks (i.e. initial, target or unsafe regions) make it hard to learn a good RL policy. For instance, in Figure 1, the initial `MID` region of the `MID-GOAL` edge in the abstract graph is under-specified, as it overlaps with a trap state. The trap state is not immediately obvious as there is a path from `MID` to `GOAL`. By analyzing explored traces, AUTOSPEC identifies problematic start states in `MID` and refines the region to exclude the trap (as shown in Figure 1), thereby refining the logical reach-avoid specification associated to the `MID-GOAL` edge.

We prove that all refinements maintain soundness, where satisfaction of the refined specification implies satisfaction of the original. AUTOSPEC integrates with existing SpectRL-compatible algorithms as demonstrated with DIRL Jothimurugan et al. (2021) and LSTS Shukla et al. (2024).

Our contributions:

1. A framework for automated refinement of logical RL specifications with four refinement procedures, all with formal soundness guarantees (Section 3).
2. Integration with existing specification-guided RL algorithms, enabling them to solve tasks with coarse specifications (Section 3).
3. Empirical demonstration that AUTOSPEC enables learning from specifications that existing methods cannot handle (Section 4).

**Related work.** Recent years have seen substantial progress in solving RL tasks specified via logical specifications Aksaray et al. (2016); Li et al. (2017b); Icarte et al. (2018); Camacho et al. (2019); Giacomo et al. (2019); Hasanbeig et al. (2022; 2019); Hahn et al. (2019); Jothimurugan et al. (2019; 2021); Xu & Topcu (2019). Many works consider different fragments of Linear Temporal Logic (LTL) or their variants for specifying RL tasks. Icarte et al. (2018); Camacho et al. (2019) consider tasks that can be specified using deterministic finite automata (DFA) and solve them by *reward machines*, which decompose these tasks and translate them into a reward function. The reward function can then be used to train existing RL algorithms. Li et al. (2017b) considers a variant of

LTL called TLTL for specifying tasks and propose a method for translating these specifications into continuous reward functions. Hasanbeig et al. (2022; 2019); Hahn et al. (2019) study the translation of tasks specified in LTL into reward functions. Alur et al. (2022) examines the theoretical questions related to the translation of logical specifications into reward functions.

Jothimurugan et al. (2019) defines the specification language SpectRL, a finitary fragment of LTL and provides justification for using this language to define specifications for RL tasks. A compositional method that decomposes SpectRL specifications into an abstract graph and constructs a reward function for each abstract graph edge was proposed in Jothimurugan et al. (2021). The approach by Toro Icarte et al. (2019) focuses on discovering optimal reward structures through environmental exploration and reward analysis. Compositional methods are further explored by Neary et al. (2022), who propose removing unfulfillable subtasks, and Neary et al. (2023), who introduce verification techniques to certify learned policies. Zikelic et al. (2023b) propose CLAPS, a compositional method for learning neural network policies with formal guarantees on the satisfaction of SpectRL specifications, thus advancing the applicability to safety-critical RL applications by utilizing prior methods for learning reach-avoid policies with formal guarantees Lechner et al. (2022); Zikelic et al. (2023a); Chatterjee et al. (2023).

Recent advancements also include LTL2Action Vaezipoor et al. (2021), which translates LTL specifications into sequences of tasks for RL agents. Other recent approaches, such as Qiu et al. (2023) and DeepLTL Jackermeier & Abate (2025), leverage goal-conditioned RL and automata-based architectures to solve complex LTL and $\omega$-regular tasks zero-shot or in multi-task settings. Trainify Jin et al. (2022) employs counterexample-guided abstraction and refinement (CEGAR) to iteratively improve policies by addressing failure cases identified through counterexamples. However, these works primarily focus on learning policies for fixed, well-defined specifications. In contrast, AUTOSPEC studies the problem of automated logical specification refinement towards improving reward functions obtained by translation from coarse logical RL specifications. Thus, our work is complementary to the works on RL from logical specifications and can be integrated into off-the-shelf specification-guided RL algorithms to improve the performance of learned agents.

## 2 PRELIMINARIES

**MDPs.** A Markov Decision Process (MDP) is a tuple $M = (S, A, P, R, \gamma)$, where $S \subseteq \mathbb{R}^n$ is the state space, $A \subseteq \mathbb{R}^m$ is the action space, $P : S \times A \times S \to [0,1]$ is the probabilistic transition function, $R$ is the (possibly non-Markovian) reward function, and $\gamma$ is the discount factor. Let $\eta : S \to [0,1]$ be the initial state distribution. A trajectory $\zeta$ in $M$ is a sequence of states and actions $\zeta = s_0, a_0, s_1, a_1, \ldots$ where $s_i \in S$ and $a_i \in A$. We use $\mathcal{Z}$ to denote the set of all trajectories in $M$ and $\mathcal{Z}_f$ to denote the set of all finite trajectories in $M$, which are finite prefixes of trajectories ending in states. A (pure) policy $\pi : \mathcal{Z}_f \to A$ assigns an action to each finite trajectory, and a non-Markovian reward $R : \mathcal{Z}_f \to \mathbb{R}$ assigns a reward to a finite trajectory. The MDP $M$ under any policy $\pi$ gives rise to a probability space over the set of all trajectories in the MDP Puterman (1994). We use $\mathbb{P}^\pi$ and $\mathbb{E}^\pi$ to denote the probability measure and the expectation operator in this probability space, respectively.

**Logical specifications for Reinforcement Learning.** In this work, we are solving RL tasks defined by logical specifications. Formally, a *logical specification* (or, simply, a *specification*) is a boolean function $\phi : \mathcal{Z} \to \{\text{true}, \text{false}\}$ which specifies whether a trajectory in the MDP satisfies the specification. We write $\zeta \models \phi$ whenever a trajectory $\zeta$ satisfies the specification $\phi$. The objective of a specification-guided RL task is to find a policy $\pi^*$ that maximizes the probability of satisfying the given specification $\phi$, i.e. $\pi^* \in argmax_\pi \mathbb{P}^\pi[\zeta \models \phi]$. Specification-guided RL algorithms use the specification to create a dense reward that guides the policy search, and therefore outperform algorithms that cannot leverage the specification for learning and instead require manual reward engineering Jothimurugan et al. (2021); Shukla et al. (2024).

**SpectRL specification logic.** We consider RL tasks specified in the SpectRL specification logic. SpectRL Jothimurugan et al. (2019) is a fragment of Linear Temporal Logic (LTL) which consists of all boolean and sequential combinations of reach-avoid tasks. Formally, a specification in SpectRL is defined in terms of *predicates* and *specification formulas*. An atomic predicate is a function $a : S \to \{\text{true}, \text{false}\}$ which defines a set of states that satisfy the atomic predicate. A predicate is a boolean combination of atomic predicates, i.e. $b := a \mid b_1 \wedge b_2 \mid b_1 \vee b_2$, where $a$ is an atomic

predicate and $b_1$ and $b_2$ are predicates. Specification formulas in SpectRL are defined by the grammar

$$\phi := \text{achieve } b \mid \phi_1 \text{ ensuring } b \mid \phi_1 ; \phi_2 \mid \phi_1 \text{ or } \phi_2 \tag{1}$$

where $b$ is a predicate and $\phi_1$ and $\phi_2$ are specification formulas. Intuitively, "achieve $b$" requires the agent to reach a state in which the predicate $b$ is satisfied. The clause "$\phi_1$ ensuring $b$" requires the agent to satisfy the specification $\phi$ while only visiting states in which the predicate $b$ is satisfied. The clause "$\phi_1 ; \phi_2$" requires the agent to first satisfy specification $\phi_1$ and then satisfy specification $\phi_2$. The clause "$\phi_1$ or $\phi_2$" requires satisfaction of at least one of $\phi_1$ or $\phi_2$. See Jothimurugan et al. (2019) for the formal definition of the semantics of each clause.

**Abstract graphs for SpectRL specifications.** It was shown in Jothimurugan et al. (2021) that each specification written in the SpectRL specification logic can be translated into an equivalent abstract graph. An *abstract graph* is a directed acyclic graph (DAG) whose vertices represent sets of MDP states and whose edges are annotated with sets of safe MDP states. Hence, each abstract graph edge defines a *reach-avoid specification*, where the task is to reach the set of states defined by the target vertex of the edge starting from the set of states defined by the source vertex of the edge, while staying within the set of safe states defined by the edge.

**Definition 1** (Abstract graph). *An* abstract graph $G = (V, E, \beta, s, t)$ *is a DAG, where $V$ is a finite set of vertices, $E$ is a finite set of edges, $\beta : V \cup E \to \mathcal{B}(S)$ is a labeling function that maps each vertex and each edge to a subset of the MDP states $S$, $s \in V$ is the source vertex and $t \in V$ is the target vertex. Furthermore, we require that $\beta(s) = support(\eta)$ is the support of the initial state distribution $\eta$ of the MDP.*

Given a trajectory $\zeta$ in the MDP and an abstract graph $G = (V, E, \beta, s, t)$, we say that $\zeta$ satisfies *abstract reachability* for $G$ (written $\zeta \models G$) if it gives rise to a path in $G$ that traverses $G$ from $s$ to $t$ and satisfies the reach-avoid specifications of every traversed edge. It was shown in Jothimurugan et al. (2021) that, given any SpectRL specification $\phi$, one can construct an abstract graph $G$ such that $\zeta \models \phi$ if and only if $\zeta \models G$ holds for each trajectory $\zeta$ in the MDP. Hence, solving an RL task for a SpectRL specification reduces to solving an abstract reachability task in the abstract graph $G$.

**Problem statement.** Given an MDP $M$ and a SpectRL specification $\phi$, our goal is to learn a policy $\pi$ such that the probability $\mathbb{P}^\pi[\zeta \models \phi]$ of a trajectory satisfying the specification is maximized.

**Specification refinement.** In order to solve this problem, we will utilize a common approach in specification-guided RL, to first translate the logical specification $\phi$ to a (non-sparse) reward function and then learn a policy by using existing RL algorithms with this reward function. However, if the probability of the specification being satisfied under the learned policy is unsatisfactory (i.e. below some desired probability threshold $p \in [0, 1]$), we will then refine the logical specification $\phi$ into a new SpectRL specification $\phi_r$. We will then repeat the above process until the probability of the specification being satisfied under the learned policy becomes satisfactory.

**Definition 2** (Specification refinement). *Given two logical specifications $\phi$ and $\phi_r$, we say that $\phi_r$ refines $\phi$, if any MDP trajectory that satisfies the refined specification $\phi_r$ also satisfies the specification $\phi$. That is, if for an MDP trajectory $\zeta$ we have $(\zeta \models \phi_r) \implies (\zeta \models \phi)$.*

## 3 Automated Refinement of RL Specifications

We now present AUTOSPEC, a framework for automated refinement of logical specifications in RL tasks. The key insight is that specification failures often stem from identifiable issues that can be systematically addressed: overly broad target regions, insufficient safety constraints, missing waypoints, or lack of alternative paths. When a specification-guided RL algorithm $\mathcal{A}$ fails to learn a satisfactory policy for specification $\phi$, AUTOSPEC identifies which components caused the failure and applies targeted refinements to improve learnability while maintaining soundness.

AUTOSPEC operates as a wrapper around any SpectRL-compatible algorithm. It monitors the learning process, and when a policy $\pi$ fails to satisfy the specification with probability at least $p \in [0, 1]$ (a user-provided threshold), it computes a refined specification $\phi_r$ such that: (1) satisfaction of $\phi_r$ implies satisfaction of $\phi$ (soundness), and (2) $\phi_r$ provides additional structure that makes it easier to learn. This refined specification is returned to algorithm $\mathcal{A}$ to continue learning. Through this iterative refinement process, AUTOSPEC enables solving RL tasks with coarse specifications that would otherwise be unlearnable.

**Overview of AUTOSPEC.** Algorithm 1 shows the complete AUTOSPEC framework. The algorithm takes as input an MDP $M$, a SpectRL specification $\phi$, a satisfaction threshold $p$, and any specification-guided RL algorithm $\mathcal{A}$. It first translates $\phi$ into an abstract graph $G$ and uses $\mathcal{A}$ to learn policies for the graph edges. For each edge $e$ where $\mathcal{A}$ learned a policy but that policy fails to achieve satisfaction probability $p$, AUTOSPEC applies four refinement procedures in sequence: SeqRefine, AddRefine, PastRefine, and OrRefine. This ordering reflects increasing levels of structural modification; from local predicate adjustments to graph topology changes. The first refinement that successfully improves performance above threshold $p$ is applied, the graph is updated, and policies are relearned before proceeding to the next edge.

---

**Algorithm 1** AUTOSPEC

---

**Require:** MDP $M$, specification $\phi$, threshold $p \in [0, 1]$, spec-guided RL algorithm $\mathcal{A}$
  $G \leftarrow$ abstract graph corresponding to $\phi$
  $\Pi \leftarrow \mathcal{A}(G)$ [set of policies for edges in $G$ learned by algorithm $\mathcal{A}$]
  **for** $e = u \rightarrow u'$ an edge in $G$ **do**
    $\pi_e \in \Pi \leftarrow$ policy learned for edge $e$ (Null if $\mathcal{A}$ does not learn a policy for edge $e$)
    **if** $\pi_e$ is not Null and $\mathbb{P}(\pi_e) < p$ **then**
      $\zeta \leftarrow$ sampled trajectories of the system
      **if** LEARNPOLICY($e$,SEQREFINE($e$,$G$,$\zeta$))$> p$ **then**
        $G \leftarrow$ SEQREFINE($e$,$G$,$\zeta$)
      **else if** LEARNPOLICY($e$,ADDREFINE($e$,$G$,$\zeta$))$> p$ **then**
        $G \leftarrow$ ADDREFINE($e$,$G$,$\zeta$)
      **else if** LEARNPOLICY($e$,PASTREFINE($e$,$G$,$\zeta$))$> p$ **then**
        $G \leftarrow$ PASTREFINE($e$,$G$,$\zeta$)
      **else if** LEARNPOLICY($e$,ORREFINE($e$,$G$,$\zeta$))$> p$ **then**
        $G \leftarrow$ ORREFINE($e$,$G$,$\zeta$)
      **end if**
      $\Pi \leftarrow \mathcal{A}(G)$ [set of policies for updated abstract graph $G$]
    **end if**
  **end for**
  **Return** $G$ and $\Pi$

---

AUTOSPEC iterates through all edges $e = u \rightarrow u'$ of the abstract graph $G$ for which the specification-guided RL algorithm $\mathcal{A}$ has learned a policy but for which the reach-avoid task satisfaction probability is below the provided probability threshold $p$. For each such edge, AUTOSPEC performs four refinement procedures that focus on different possible reasons for the edge $e = u \rightarrow u'$ being challenging for learning a satisfactory policy.

SeqRefine, which is invoked first, tries to locally refine the problematic edge $e = u \rightarrow u'$ by using predicate refinement techniques to refine the labeling function at the target region associated to the vertex $u'$ and the safety region associated to the edge $e$. If SeqRefine fails to improve performance above threshold, AUTOSPEC invokes AddRefine which attempts to add a waypoint (i.e. a new abstract graph vertex) between the vertices $u$ and $u'$, making path-finding easier. If AddRefine also fails, AUTOSPEC invokes PastRefine which tries to refine the source node $u$. Finally, if other refinement procedures fail, AUTOSPEC invokes OrRefine which aims to find alternative paths to $u'$.

After each attempted refinement, an off-the-shelf RL algorithm (LEARNPOLICY in Algorithm 1) is used to estimate the satisfaction probability of the refined edge. When a refinement succeeds in achieving satisfaction probability above $p$, the refined abstract graph $G$ is updated and AUTOSPEC applies the specification-guided RL algorithm $\mathcal{A}$ to learn a new set of edge policies $\Pi$ for the entire graph. At the end, the final abstract graph $G$ corresponds to the refined specification $\phi_r$ of the input specification $\phi$.

## 3.1 SPECIFICATION REFINEMENT SUBPROCEDURES

We now define the four specification refinement subprocedures used by AUTOSPEC in Algorithm 1. Each procedure addresses a specific type of specification inadequacy, and they are applied in order of increasing structural modification to the abstract graph. The detailed pseudocodes are provided in the Appendix. Once a problematic edge $e = u \rightarrow u'$ is identified (an edge with satisfaction probability

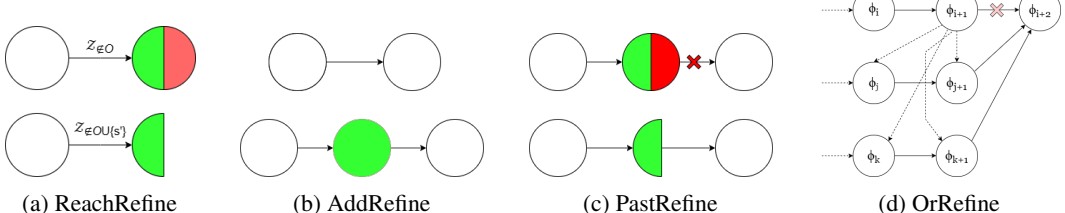

(a) ReachRefine      (b) AddRefine      (c) PastRefine      (d) OrRefine

Figure 2: Illustrations of abstract graph refinement processes. (a) ReachRefine demonstrating removal of failed part of goal region and addition of unsafe states set to existing avoid. (b) AddRefine demonstrating addition of waypoint between 2 nodes. (c) PastRefine removes part of the source node from where the agent failed (red) to get to the target, and keeps successful start states (green). (d) OrRefine shows how alternative paths (dotted lines) to target are constructed using existing specification nodes.

below threshold $p$), AUTOSPEC samples trajectories $\zeta$ using the learned policy, where the number of trajectories is an algorithm hyperparameter.

**SeqRefine: Refining Predicates.** The first refinement subprocedure addresses overly coarse predicates in the reach and avoid conditions. For edge $e = u \to u'$, SeqRefine refines both the target predicate $b = \beta(u')$ and the safety predicate $c = \beta(e)$ by calling two subprocedures:

*ReachRefine* collects all states along sampled trajectories that successfully reached the goal region $b$. The refined goal region is computed as $b_r = b \cap$ ConvexHull(reached states), effectively excluding unreachable portions of the original target region. *AvoidRefine* collects states where trajectories entered unsafe regions (complement of $c$). The refined safe region is computed as $c_r = c \setminus$ ConvexHull(last $k$ unsafe trajectory states), where $k$ is a hyperparameter specifying how much of the trajectory tail to remove. This removes demonstrated unsafe areas from the safety region.

SeqRefine returns a refined graph $G_r$ identical to $G$ except with updated labeling: $\beta_r(u') = b_r$ and $\beta_r(e) = c_r$. This refinement provides more precise guidance by excluding problematic regions discovered through exploration.

**AddRefine: Introducing Waypoints.** The second refinement addresses long or complex paths by decomposing them. When direct navigation from $u$ to $u'$ proves difficult, AddRefine introduces an intermediate vertex $u''$ by collecting midpoint states from successful trajectories that reached $\beta(u')$, defining $\beta(u'') = \beta(e) \cap$ ConvexHull(midpoints), and replacing edge $e = u \to u'$ with two edges: $e'' = u \to u''$ and $e' = u'' \to u'$. This decomposition breaks a challenging long-horizon task into two shorter subtasks that are easier to learn.

**PastRefine: Partitioning Source Regions.** The third refinement addresses heterogeneous starting conditions where some initial states in $u$ consistently lead to success while others lead to failure. PastRefine separates trajectories into successful and failing sets based on whether they satisfied edge $e$, then learns a hyperplane separating successful from failing initial states. It creates region $b_r$ containing successful starting states and introduces new vertex $u^*$ with $\beta(u^*) = b_r$ having the same incoming edges as $u$. The refinement replaces problematic edge $e = u \to u'$ with $e^* = u^* \to u'$. As shown in Figure 2(a), this refinement identifies and isolates promising initial conditions while preserving the original vertex $u$ and its connections.

**OrRefine: Exploiting Alternative Paths.** The fourth refinement addresses blocked or infeasible direct paths by leveraging the existing graph structure. When the path through edge $e = u \to u'$ cannot be made satisfactory, OrRefine identifies alternative parents of $u'$ (vertices $u_i$ with existing edges $e_i = u_i \to u'$), and for each viable $u_i$, adds new edge $e_{new} = u \to u_i$ with $\beta(e_{new}) = \beta(e)$ and $\beta(e_i) = \beta(e) \cap \beta(e_i)$. It then tests if the alternative path $u \to u_i \to u'$ achieves the threshold. As illustrated in Figure 2(b), this creates alternative routes to the target using only existing vertices, maintaining all original safety constraints. OrRefine can iteratively explore ancestors of $u_i$ if the direct connection fails.

As shown in Algorithm 1, any specification-guided RL algorithm that is applicable to SpectRL specifications and that learns policies for edges in the abstract graph can be integrated into the AUTOSPEC framework. The specification-guided RL algorithm learns policies for edges in the

abstract graph, until it is unable to proceed beyond an edge with a sufficient satisfaction probability. We then perform the refinements in AUTOSPEC, using sampling to estimate the satisfaction probability of each refinement until one is found to exceed the threshold. This refinement is used to create an updated abstract graph and an updated set of edge policies are learned with respect to this graph.

## 3.2 CORRECTNESS OF AUTOSPEC

The following theorem establishes correctness of AUTOSPEC, showing that the specification $\phi_r$ computed by AUTOSPEC is indeed a refinement of the input specification $\phi$. The proof, provided in the Appendix, proceeds by proving that each of the four refinement procedures results in a specification refinement.

**Theorem 1** (Correctness of AUTOSPEC). *Given an abstract graph $G$ of a SpectRL specification $\phi$ and an edge $e$, AUTOSPEC computes a specification $\phi_r$ and returns an abstract graph $G_r$ and an edge $e_r$ such that $\phi_r$ refines $\phi$. That is, for any MDP trajectory $\zeta$, we have $(\zeta \models \phi_r) \implies (\zeta \models \phi)$.*

**Incompleteness of the Specification Refinement Problem.** AUTOSPEC provides *soundness* guarantees – as shown in Theorem 1, the produced specification is *guaranteed* to be a refinement of the original specification as in Definition 2, and every trajectory that satisfies the refined satisfaction must also satisfy the original specification. However, AUTOSPEC does not provide completeness guarantees. This is not a limitation of AUTOSPEC, but an inherent property of the specification refinement problem itself because the problem is *undecidable*. This is because, even for the simplest case of reachability specifications (e.g., "reach region $G$ with probability at least $p$"), deciding whether a given policy satisfies a specification is *undecidable* for general continuous-state MDPs or probabilistic programs capable of encoding Turing-complete behavior Kaminski & Katoen (2015). Consequently, no specification refinement algorithm can be both sound and complete. AUTOSPEC therefore focuses on soundness, which is important towards ensuring that a policy for the refined task also solves the task defined by the original specification.

## 4 EXPERIMENTAL EVALUATION

We evaluate AUTOSPEC on its ability to diagnose and repair specification failures that prevent existing algorithms from learning satisfactory policies. Our experiments address three questions: (1) Can AUTOSPEC correctly identify which refinement type is needed for different failure modes? (2) Do the refinements enable learning from previously unlearnable specifications? (3) What are the requirements and limitations of the refinement process?

### 4.1 EXPERIMENTAL SETUP

We integrate AUTOSPEC with two specification-guided RL algorithms: DIRL Jothimurugan et al. (2021), which uses Dijkstra-style graph search with systematic exploration, and LSTS Shukla et al. (2024), which uses multi-armed bandits for edge selection with epsilon-greedy exploration. These algorithms differ fundamentally in their exploration strategies, allowing us to examine how AUTOSPEC's effectiveness depends on the underlying learning algorithm. We evaluate on two domains specifically chosen to stress-test different aspects of specification refinement:

**n-Rooms:** Grid-based navigation with walls and doors, providing controlled tests of specific failure modes. State space: $(x, y, \theta, d) \in \mathbb{R}^4$ (position, angle to goal, distance). Action space: $(v, \theta) \in \mathbb{R}^2$ (velocity, direction). The n-rooms domain has been extensively used in specification-guided RL research Jothimurugan et al. (2021; 2019); Zikelic et al. (2023b) as it provides clear geometric structure while still presenting challenging long-horizon tasks. Its modular room structure naturally creates the types of specification failures we aim to address: trap states at room boundaries, dangerous narrow passages between rooms, and multiple alternative paths through different door configurations.

**PandaGym Gallouédec et al. (2021):** Robotic manipulation requiring 3D navigation around obstacles. This domain tests refinement in high-dimensional continuous control where geometric intuitions may not apply directly. Following recent work showing the challenges of specification-guided RL in manipulation tasks Shukla et al. (2024), we use this domain to validate that our convex hull and hyperplane-based refinements remain effective in high-dimensional spaces where human intuition about specification failures is limited.

For learning edge policies, both algorithms use PPO Schulman et al. (2017) with stable-baselines3 Raffin et al. (2021) implementation, following the standard practice in recent specification-guided RL work Jothimurugan et al. (2021); Zikelic et al. (2023b). We use 2-layer networks (64 neurons each), learning rate 0.0003, and standard PPO hyperparameters. In all experiments we evaluate refinements using a deliberately high satisfaction threshold ($p = 0.99$). The purpose of this choice is methodological: by selecting a probability level that is difficult to achieve under coarse or under-specified predicates, we can clearly observe how the cumulative probability of satisfying the specification improves as AutoSpec performs successive refinements. Using such a stringent threshold ensures that even small improvements in guidance become visible in the satisfaction curves and allows us to measure the full extent of the benefit provided by refinement, independent of how poorly the initial specification performs.

All experiments are repeated over five random seeds. Plots report the mean across the five runs, with error bars showing the empirical mean $\pm$ variance. Specifically, each data point in the learning curves represents the performance of a policy trained for the distinct number of timesteps indicated on the x-axis (e.g., policies are trained for 80,000, 100,000, and 120,000 steps independently across 5 seeds). To estimate the specification satisfaction probability (y-axis), the trained policy for each edge is evaluated over 1000 rollout trajectories to empirically count successful versus failed attempts. The final success probabilities displayed in the plots are calculated using the product of success probability of the best path from start to goal.

## 4.2 ALGORITHM-DEPENDENT EFFECTIVENESS: DIRL VS LSTS

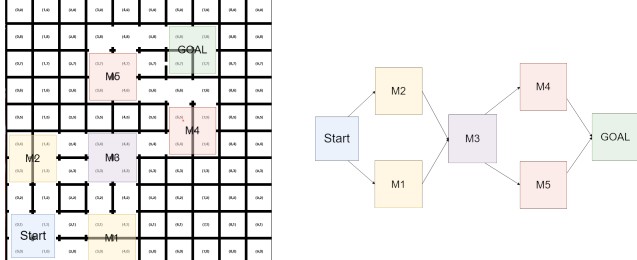

Figure 3: 100-rooms Environment with marked regions its DAG specification

Our experiments reveal that AUTOSPEC's effectiveness depends critically on the base algorithm's exploration strategy. We demonstrate this through a 100-rooms environment (Figure 3) with the complex specification: $\phi = \phi_{start}; (\phi_{m1} \text{ or } \phi_{m2}); \phi_{m3}; (\phi_{m4} \text{ or } \phi_{m5}); \phi_{goal}$

This specification structure, with multiple disjunctive branches and sequential compositions, represents the type of complex task decomposition that prior work Jothimurugan et al. (2019; 2021) has identified as necessary for real-world applications but challenging for existing algorithms. The 100-rooms scale specifically tests whether refinements remain effective when the state space is large enough that exhaustive exploration is infeasible, reflecting concerns raised in Shukla et al. (2024) about scalability of compositional methods.

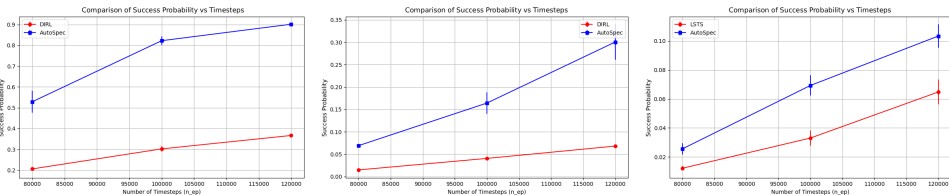

(a) Mid-goal DIRL performance  (b) Full-spec DIRL performance  (c) Mid-goal LSTS performance

Figure 4: Task satisfiability curves representing performances of DIRL and LSTS for subspecifications and complete specification

**With DiRL (Successful Refinement).** As shown in Figure 4(a-b), DiRL's systematic exploration enables successful refinement. The algorithm explores edges in order of estimated difficulty, providing sufficient trajectory data for each edge before moving to the next. AUTOSPEC successfully applies ReachRefine on the $\phi_{m1}$ edge to remove unreachable portions of the target region, PastRefine on the $\phi_{m3}$ edge to identify successful starting regions, and OrRefine when direct paths fail to find alternative routes through $\phi_{m2}$. The satisfaction probability improves from near 0% to approximately 60% through these refinements.

**With LSTS (Refinement Failure).** Figure 4(c) shows LSTS failing on the same specification. The bandit-based exploration spreads effort across all edges simultaneously, preventing deep exploration of any single edge. Consequently, edges to M4, M5, and Goal achieve 0% satisfaction, providing no successful trajectories for refinement computation. AUTOSPEC correctly reports its inability to refine without samples, demonstrating that refinement quality fundamentally depends on the base algorithm's exploration strategy.

**Evaluation on Randomized 100-rooms and predicate placement** To evaluate the generalization capabilities of our framework, we deployed AUTOSPEC in a procedurally generated 4-Way Gridworld where wall connectivity, and predicate placement were fully randomized for each seed (see the Appendix for generation details). This setup specifically tests the system's ability to synthesize policies without reliance on hand-engineered specifications or environment-specific heuristics. As shown in Figure 6, AUTOSPEC significantly outperforms the DIRL baseline, achieving a terminal success probability of approximately $60\%$ compared to the baseline's stagnation at $20\%$. These results confirm that AUTOSPEC autonomously identifies and resolves task bottlenecks, raising success rate of critical transitions from $< 20\%$ to $> 90\%$ via automatic refinement, see the Appendix.

## 4.3 EVALUATION OF INDIVIDUAL REFINEMENTS

We design targeted experiments isolating specific failure modes to validate each refinement procedure.

**SeqRefine: Trap State Elimination (Figure 7). Setup:** 9-rooms environment where the goal region includes a blocked room creating a trap state. **Failure mode:** Agent reaches the trap portion of the goal and cannot escape. **Refinement:** ReachRefine identifies that successful trajectories only reach the accessible portion of the goal. The refined specification excludes the trap region: $b_r = b \cap \text{ConvexHull}(\text{reached states})$. **Result:** Satisfaction probability improves from 15% to 85%, demonstrating AUTOSPEC's ability to learn environmental constraints not captured in the original specification.

**SeqRefine: Safety Constraint Discovery (Figure 8). Setup:** 9-rooms with a narrow dangerous passage below the goal. **Failure mode:** Shortest path goes through narrow passage where agent frequently fails. **Refinement:** AvoidRefine identifies failure states near the narrow passage. The refined specification expands the avoid region: $c_r = c \setminus \text{ConvexHull}(\text{last 10 failure states})$. **Result:** Agent learns to use wider but longer safe path, improving satisfaction from 30% to 75%.

**AddRefine: Waypoint Introduction (Figure 9). Setup:** Long-horizon navigation across multiple rooms. **Failure mode:** Direct path too complex for single policy to learn reliably. **Refinement:** AddRefine identifies midpoints of successful trajectories and introduces intermediate vertex $u''$. **Result:** Decomposes task into two manageable subtasks, improving satisfaction from 20% to 90%.

**PastRefine: Initial State Partitioning (Figure 10). Setup:** Starting region includes states from which goal is unreachable. **Failure mode:** Policy cannot succeed from certain initial states. **Refinement:** PastRefine learns hyperplane separating successful from failing starts. **Result:** Focuses learning on viable initial states, improving satisfaction from 40% to 80%.

**OrRefine: Alternative Path Discovery (Figure 11). Setup:** Specification with multiple possible paths: $\phi_{MID1}; \phi_{GOAL}$ or $\phi_{MID2}; \phi_{GOAL}$. **Failure mode:** Direct path through MID1 blocked. **Refinement:** OrRefine adds edge $\phi_{MID1} \rightarrow \phi_{MID2}$, creating alternative route. **Result:** Enables satisfaction through alternate path when direct path has 0% success.

## 4.4 HIGH-DIMENSIONAL VALIDATION: PANDAGYM

To validate beyond grid environments, we test AUTOSPEC on PandaGym's continuous 3D manipulation task. The specification requires navigating around an invisible wall:

Figure 5: Evaluation of AUTOSPEC on PandaGym: (a) Two perspectives of the environment (1st and 2nd Figures), where the red region is an intermediate goal and an invisible wall blocks direct paths. (b) Performance of DiRL with and without AUTOSPEC: ReachRefine on first edge (3rd Figure) and PastRefine on second edge (4th Figure).

($reach$ red-region avoid wall); ($reach$ green-region avoid wall). The invisible wall creates a challenging scenario where the agent cannot directly observe the obstacle, making specification refinement crucial.

As shown in Figure 5, AUTOSPEC with DiRL successfully applies ReachRefine on the first edge to identify and exclude unreachable portions of the red region behind the wall, focusing the policy on achievable subgoals. On the second edge, PastRefine learns that only certain approach angles from the red region lead to successful reaching of the green region, effectively partitioning the intermediate state space based on trajectory outcomes. This demonstrates that AUTOSPEC's geometric refinements (convex hulls for ReachRefine, hyperplanes for PastRefine) remain effective in high-dimensional spaces where human intuition about the specification failures would be difficult. The success in this domain is particularly noteworthy because the refinements must capture 3D spatial relationships without explicit knowledge of the obstacle geometry.

**Computational Overhead.** AUTOSPEC avoids full retraining by only updating the policies associated with the identified subset of refined edges $\mathcal{R}$. The total computational cost is formalized as $T_{\text{total}} = T_{\text{base}} + \sum_{e \in \mathcal{R}} T_e$. Since $|\mathcal{R}|$ is typically small relative to the initial graph size, the aggregate overhead is bounded (empirically $T_{\text{total}} \leq 2T_{\text{base}}$). In the 100-room experiments, the baseline required $\sim 240$s to evaluate the 8 fixed edges, while AUTOSPEC averaged $390 \pm 42$s. This overhead corresponds directly to the training of 4–7 additional refinement edges per seed, with the observed variance ($\sigma \approx 42$s) driven by the differing topological complexity of the randomized environments. This computational investment is highly efficient, scaling linearly with the number of detected bottlenecks rather than the global state space size. Given the substantial improvement in success probability (from $\approx 20\%$ to $\approx 60\%$), this bounded overhead represents a favorable trade-off for achieving robust autonomy in stochastic domains.

## 5 CONCLUSION

We presented AUTOSPEC, a framework for automated refinement of coarse-grained logical specifications in reinforcement learning. AUTOSPEC addresses two common specification issues — coarse formulas and coarse labeling functions through four refinement procedures that maintain formal soundness. Our experiments on n-rooms and PandaGym environments demonstrate that AUTOSPEC can improve specification satisfiability when integrated with existing algorithms like DiRL and LSTS.

Our evaluation also reveals fundamental limitations: AUTOSPEC requires sufficient exploration data from the base algorithm to compute meaningful refinements. When algorithms fail to generate successful trajectories (as LSTS did on complex specifications), refinement becomes impossible. Despite these limitations, AUTOSPEC represents the first systematic approach to automatically refining logical specifications based on learning failures. Future work should address reducing exploration requirements for refinement and extending beyond SpectRL to more expressive temporal logics, such as infinite-horizon $\omega$-regular specifications. While AUTOSPEC currently relies on finite witnesses, it could be adapted to these settings by decomposing tasks into a finite prefix (amenable to our current DAG-based refinement) and a cyclic suffix (which would require extending our witness analysis to handle infinite behaviors). The design of good specifications remains challenging in practice, and automated refinement is an important step toward making specification-guided RL more practical.

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
