# OpenReview forum: "Automating the Refinement of Reinforcement Learning Specifications"
_ICLR.cc/2026/Conference — ICLR 2026 Poster_

### Official Review · Reviewer_fNYW · 2025-10-26

**Soundness:** 3
**Presentation:** 3
**Contribution:** 3
**Rating:** 6
**Confidence:** 5

**Summary:**

This paper introduces AutoSpec, a framework designed to automatically refine logical specifications for reinforcement learning (RL) tasks. While logical specifications can guide RL agents towards complex goals, a common issue is that coarse-grained or under-specified definitions may prevent agents from learning useful policies.

The core idea of AutoSpec is to use an "exploration-guided strategy" to automatically search for a more detailed specification. This "refined" specification is stricter than the original but provides additional guidance to the RL algorithm, making the learning process easier. Crucially, the framework guarantees "soundness": any trajectory satisfying the new specification must also satisfy the original coarse one. Theoretical justifications are provided. Experiments demonstrate that agents using specifications refined by AutoSpec can solve more complex control tasks than before.

**Strengths:**

1. The issue of "specification too coarse to learn from" is a significant practical hurdle in specification-based RL. This paper tackles this problem directly, which is highly valuable.
2. The primary contribution of AutoSpec is its ability to refine specifications without human intervention. This greatly lowers the barrier to using logical specifications, which might otherwise require extensive manual tuning by domain experts. Also, theoretical justifications are provided to show the framework doesn't just modify specifications arbitrarily; it guarantees that the refined specification is a valid "subset" of the original. This is crucial.
3. The refinement processes to abstract graphs are clear and intuitive, the order of refinement are also reasonable. Overall, the algorithm is easy to follow.

**Weaknesses:**

1. I am not fully convinced by the name *SpectRL* specification logic. To me, they are just standard subset of Linear Temporal Logic (LTL), clarifying this in the paper clearly is sufficient. We do not really need a separate notation.
2. The paper mentions that AutoSpec searches for a refinement in order. How large is this search space for each environment, and what is the overhead of this additional procedure? These are not mentioned in the paper.
3. As discussed in the paper, the reliability of the AutoSpec are heavily dependent on its base algorithm. This limitation is understandable, yet it would be better if additional discussion and proposals can be provided.
4. As I mentioned in weaknesses 1, this work only discusses a subset of LTL, while other works have already proposed some insights and solutions to the problem [1, 2]. Please consider add discussion of these works and give your own insights.
5. This is minor, I noticed that two important concepts mentioned in the abstract never mentioned again in the main text. Please explain  what are "under-specified" and "exploration-guided strategy" in the main text. I know these are the summarizations of later component, but it is better to make them clear.

[1] Qiu et al. Instructing goal-conditioned reinforcement learning agents with temporal logic objectives. NeurIPS 2023.

[2] Jackermeier and Abate, DeepLTL: Learning to Efficiently Satisfy Complex LTL Specifications for Multi-Task RL. ICLR 2025.

**Questions:**

1. See weaknesses 2, please explain the search space and overhead of AutoSpec.
2. See weaknesses 3, please explain the difficulty of refinement in AutoSpec.
3. See weaknesses 4, can you discuss [1] and [2] in the paper, and provide some insights on whether AutoSpec can be applied to $ \omega $-regular LTL specifications?
4. It seems that AutoSpec detect the failures of a policy and perform refinement, this is good. However, it is possible to perform "active" refinement rather than "passive"? This could be very interesting.

---

> ### Author Response · Authors · 2025-11-21
>
> We thank the reviewer for their positive assessment and for highlighting the practical value of our work. We have incorporated your suggestions regarding related work and definitions to improve the manuscript.
>
> ### **A. Address Questions**
>
> **1. Search Space & Computational Overhead (Question 1 & Weakness 2)**
>
> * **Search Space:** AutoSpec does not search an unbounded combinatorial space. The search space is **linear** in the number of edges of the abstract graph ($O(|E|)$). We iterate through edges, checking if the success probability is below the threshold $p$. If it is, we trigger the fixed sequence of 4 refinement operators (Choose 1st that satisfies threshold). This makes the search process itself negligible in cost compared to training.
> * **Computational Overhead:** The dominant cost is retraining the policy. We added a new subsection **"Computational Overhead" (Section 4.4)** explicitly quantifying this. The total cost is approximately $T_{total} \approx T_{base} + T_{refine}$, incurring a **~2x** overhead compared to a standard training run.
> * **Empirical Data:** In the 100-rooms domain, the baseline training took **~240s**, while the complete AutoSpec loop averaged **390s**, to yield a 3x improvement in task satisfaction. We view this as an efficient trade-off for automating the manual specification debugging loop.
>
> **2. Dependence on Base Algorithm (Question 2 & Weakness 3)**
>
> * **Response:** This is correct. AutoSpec is designed to be **sound** (Theorem 1), meaning we never hallucinate a refinement; we only refine based on *empirical witnesses* (successful trajectories). If the base algorithm (e.g., LSTS) yields zero successful traces, we cannot guarantee that a refinement is valid, so we do not act.
> * **Theoretical Context:** We view this not as a flaw but as a fundamental property of sound refinement. As discussed in our new **"Incompleteness" paragraph (Section 3.2)**, the general refinement problem is undecidable; therefore, AutoSpec's reliance on empirical witnesses is necessary for soundness.
>
> **Comparison to Qiu et al. (2023) & DeepLTL (2025) (Question 3 & Weakness 4)**
>
> * **Distinction:** We have added a detailed discussion of these works in **Section 1 (Related Work)**. We clarify that Qiu et al. and DeepLTL solve the **Control Problem**: given a *fixed, correct* specification, how do we learn a policy (often zero-shot)? AutoSpec solves the upstream **Refinement Problem**: what if the provided specification is *too coarse*?
> * **Synergy:** AutoSpec is complementary. If DeepLTL fails to solve a task because a "Reach Region" geometrically contains a trap state, AutoSpec can be used to refine that region *before* passing the corrected specification to the DeepLTL solver.
> * **$\omega$-Regular Insight:** In our revised **Conclusion**, we outline a path to applicability: Any $\omega$-regular task can be decomposed into a finite **prefix** (reaching a cycle) and an infinite **suffix** (staying in the cycle). AutoSpec's current DAG-based refinement works perfectly for the prefix. For the suffix, we propose extending our witness analysis to "lasso" traces - finding the geometric intersection of states visited during the repeated cycle.
>
> **Active vs. Passive Refinement (Question 4)**
> * **Response:** This is a fascinating direction. While AutoSpec currently uses a **passive** strategy to strictly preserve soundness, we agree that an **active** approach is the logical next step. A future framework could employ a **dynamic operator** trained to infer the specific *type* of failure from live trajectories (e.g., detecting a "bottleneck" vs. "dead-end") and apply the corresponding refinement (e.g., `AddRefine` vs. `PastRefine`) on-the-fly. This would shift the paradigm from "refining after failure" to "refining during exploration," potentially accelerating convergence significantly. A key challenge for this direction would be designing these dynamic interventions to strictly maintain **soundness guarantees**.
>
> ### **2. Respond to Weaknesses**
>
> **SpectRL vs. LTL (Weakness 1)**
>
> SpectRL is a structured fragment of LTL rather than a distinct logic.
> * **Clarification:** We use the term to refer specifically to DAG-based syntax introduced by Jothimurugan et al. (2019), which provides the structural properties (nodes as subgoals, edges as constraints) necessary for our refinement operators.
> * **Revision:** We have explicitly clarified in **Section 2 (Preliminaries)** and **Section 1 (Related Work)** that SpectRL is a "finitary fragment of Linear Temporal Logic (LTL)" to avoid overclaiming novelty on the logic itself.
>
> **Definitions of Terms (Weakness 5)**
>
> * **Revision:** We have added a discussion (Introduction L70-80 ) that provides details about "under-specified" (coarse predicates leading to local optima) and "exploration-guided strategy" (using empirical rollouts to repair logic), linking them to the trap-state example in **Figure 1**.

---

### Official Review · Reviewer_w6NH · 2025-10-31

**Soundness:** 3
**Presentation:** 2
**Contribution:** 3
**Rating:** 8
**Confidence:** 4

**Summary:**

The paper considers the problem of automatically refining logical specifications in order to help specification guided reinforcement learning algorithms. The main intuition is that when the specification is very coarse, these algorithms find it hard to learn effective policies. So they propose identifying problematic specifications and refining them to help the algorithms converge faster and also help with guided exploration.

This work uses the SpectRL specification logic which can be represented as a graph that captures different ways to satisfy the specification. They present several types of refinement procedures that modify the graph and these procedures consist of refining goal/target regions, or adding additional intermediate target regions.

In their experiments they show that their method greatly helps specification guided RL algorithms to learn effective policies in large gridworld environments as well as robotic manipulation task with obstacles.

**Strengths:**

This is one of the first few works to consider the problem of automatic refinement of RL specifications based on collected feedback from the training of policies. This is a fundamental issue because if the specification is too coarse grained then algorithms would find it hard to effectively explore the state space and learn good policies.

The problem they consider is studied in depth and many refinement techniques are proposed that are sound. The benchmarks they consider are also interesting. This work also opens up many interesting related directions that can be explored.

**Weaknesses:**

While the contributions of the paper are substantive, they can be presented better. The introduction can be expanded to give further intuition with respect to the problem being solved. Specifically, the notion of abstract graph is never introduced informally even though it is central to the paper. Perhaps it would be helpful to take the example in figure 1, and present in some detail about what the refinement procedures would produce and how it would make the learning task easier. Similarly, logical specifications for RL are also never introduced.

The related work section can also be organized better into paragraphs.

**Questions:**

1. Are there possible failure modes where the refinement procedure would follow a wrong chain of refinements that make the learning task much harder? Perhaps this deserves a short discussion?

2. In the current algorithms, the different refinements are applied in a specific order. Do you imagine situations where this order can be detrimental?

3. Why or why not dynamically choose which refinement to apply at each step?

---

> ### Author Response · Authors · 2025-11-21
> **Rebuttal**
>
> We thank the reviewer for their positive assessment of our work and for highlighting the importance of automated refinement in making specification-guided RL practical. We have incorporated your suggestions to improve the presentation and flow of the paper.
>
> ### **1. Address Questions**
>
> **Failure Modes (Question 1)**
>
> * **Soundness vs. Feasibility:** AutoSpec is designed to be **sound**, meaning every refined specification is a strict logical subset of the original. We acknowledge this can theoretically render a task impossible. For instance, consider a scenario with a narrow door where a policy succeeds 5% of the time but fails 95% (hitting the wall). `AvoidRefine` might identify the high density of failure states near the door and expand the unsafe region to cover the passage completely, blocking the path to the goal.
> * **Why we accept this risk:** While this refinement prevents task completion, it ensures that if a trajectory satisfies the refined specification, it will also satisfy the original specification. If we were to conversely *relax* the unsafe region to make the task easier, a trajectory could violate the original specification while satisfying the refinement, breaking soundness. Thus, we prioritize safety over completion.
> * **Empirical Robustness:** Despite this theoretical risk, empirically we observe that refinements guide the agent effectively. In our new **Randomized 100-rooms experiment** (Appendix A.3), AutoSpec improved task satisfaction by **3x** (from ~20% to ~60%) despite the random topology.
> * **Environment Limitations:** We also acknowledge that refinements cannot solve fundamentally impossible tasks defined by dynamics. For example, in a *Start-Mid-Goal* task, if a wall blocks the transition between the arrival point in *Mid* (bottom-half) and the required departure point for *Goal* (top-half), no refinement of the edges will yield success. This is a limitation of the environment, not the refinement logic.
>
> **Refinement Order (Question 2)**
>
> * **Heuristic Rationale:** Our heuristic prioritizes the **cost of graph modification and retraining**:
>     1.  `SeqRefine`: Modifies only the predicate labels of the current edge (Lowest cost).
>     2.  `AddRefine`: Adds a new node on the *same* edge path, requiring training for 2 sub-policies.
>     3.  `PastRefine`: Refines the current start node, which necessitates retraining two edges in the original structure.
>     4.  `OrRefine`: Searches for completely different paths, making it the most expensive operation.
> * **Safeguards:** While this heuristic could theoretically be suboptimal in specific instances, we mitigate this by only accepting a refinement if it exceeds the probability threshold (or selecting the best available candidate if none do). Given the undecidability of determining the optimal refinement order a priori, we found this cost-based heuristic to be empirically robust.
>
> **Dynamic Selection (Question 3)**
>
> * **Response:** We agree that dynamically selecting the refinement operator is an excellent idea, as it could reduce the total number of refinement steps. The key challenge is identifying specific failure patterns that map reliably to specific refinement types. At this stage, we have not identified sufficiently general patterns to construct such a dynamic operator. However, given the undecidable nature of the problem, developing learned heuristics (e.g., via meta-learning) to predict the optimal refinement type is a promising direction for future research.
>
> ### **2. Respond to Weaknesses**
>
> **Intuitive Introduction & Abstract Graphs**
>
> * **Revision:** We have added a discussion (Introduction L70-80) that provides the requested informal walkthrough. We explicitly define what makes a specification *"under-specified"* and use the **Figure 1** example to explain how the abstract graph acts as a high-level map of subgoals.
> * **Formal Definitions**
> The foundational concepts, **Logical Specifications for Reinforcement Learning** and **Abstracts Graphs for SpectRL Specifications**, are formally defined in Section 2 (Preliminaries). This structure guides the reader through understanding logical specifications for RL, the nature of SpectRL specification logic, and the subsequent process of translating SpectRL specifications into abstract graphs for compositional learning.
>
> **Related Work Organization**
>
>  **Revision:** We have reorganized the **Related Work (Pages 2-3)** into three distinct thematic paragraphs:
> 1.  **General Logical Specifications:** Reward Machines and automata-based RL.
> 2.  **SpectRL & Compositional Methods:** The specific logic and graph-based approaches our work builds upon.
> 3.  **Recent Advanced Approaches:** Discussions of recent works to position AutoSpec distinctively as a *refinement* framework rather than a zero-shot instruction framework.

---

> > ### Comment · Reviewer_w6NH · 2025-11-24
> >
> > Thanks for the response, I have no further questions/concerns.

---

### Official Review · Reviewer_Ui5g · 2025-10-31

**Soundness:** 2
**Presentation:** 2
**Contribution:** 2
**Rating:** 4
**Confidence:** 5

**Summary:**

Specification guided reinforcement learning often fails when initial logical task descriptions and their labeling functions are too coarse. The paper proposes $\mathrm{AutoSpec}$, a framework that refines SpectRL specifications through an exploration driven loop and four refinement procedures, while guaranteeing that any trajectory that satisfies the refined specification also satisfies the original specification. The method integrates with existing algorithms and is demonstrated with $\mathrm{DIRL}$ and $\mathrm{LSTS}$, where refinements help recover learnability on tasks that were previously hard to solve.

**Strengths:**

- The paper addresses an important challenge in formulating specifications for RL. Automatically refining predicates and specifications provided to the agent is a promising direction, particularly because crafting appropriate predicates is difficult and loosely defined specifications can be hard to satisfy.
- The framework is integrated with established specification guided algorithms and the experiments illustrate how the refinements interact with the different exploration strategies of $\mathrm{DIRL}$ and $\mathrm{LSTS}$.

**Weaknesses:**

- The empirical scope is narrow. Only two domains are considered, n Rooms and PandaGym. This limits the evidence for scalability and diversity of specifications. A broader study that samples many predicate regions or includes a less contrived multi room world would strengthen the case.
    - The specifications tested upon are quite limited (only 1 or 2 refinements per specification needed). A sample driven testing approach (say randomly chosen predicate regions) or a less contrived 100 room example would be more convincing of the scalability of the approach.
    - I appreciate the carefully chosen experiments for an intuition of what is happening, but some further generalization studies would help (e.g. more than 2 refinements needed and whether $\mathrm{AutoSpec}$ covers the search space appropriately).
- When there are no successful samples, certain refinements cannot be computed, as observed for $\mathrm{LSTS}$ on the complex specification. The approach is sound but not complete and it may fail to find a refinement even if one exists.
- `AvoidRefine` only enlarges the avoid set or equivalently reduces the safe set, without permitting relaxations when the avoid region is overly conservative. Algorithm 3 defines the refined safe region by removing the convex hull of recent failure states, which can bias the learner away from potentially optimal paths if the initial avoid labeling is narrow or misaligned.  This is acceptable in most situations,  but the onus is on the user specifying the initial predicate regions to start with conservative definitions. A discussion of when to relax an avoid constraint would be valuable.

**Questions:**

1. What is the computational overhead of $\mathrm{AutoSpec}$ in the reported settings, relative to running $\mathrm{DIRL}$  or $\mathrm{LSTS}$  alone? A wall clock comparison and a complexity view in terms of the number of edges and sampled trajectories per refinement would help readers assess practical costs.
2. How do the procedures behave when a specification needs several consecutive refinements? Is there an observed depth beyond which refinements fail to improve satisfaction probability or become unstable?

---

> ### Author Response · Authors · 2025-11-21
> **Rebuttal**
>
> We thank the reviewer for their constructive feedback and for recognizing the importance of automated refinement in specification-guided RL. Based on your valuable suggestion regarding empirical scope, we have significantly expanded our evaluation to include randomized domains and provided concrete runtime analysis.
>
> ### **A. Address Questions**
>
> **1. Computational Overhead & Wall Clock Comparison**
>
> * **Response:** We added a new subsection **"Computational Overhead" (Section 4.4)**. We explicitly formalize the cost as $T_{total} = T_{base} + \sum_{e \in \mathcal{R}} T_{e}$, where $\mathcal{R}$ is the subset of refined edges.
> * **Empirical Data:** We report concrete numbers from the 100-rooms environment. The baseline (original 8 edges using DIRL) required **~240s**, while the complete AutoSpec loop averaged **390s $\pm$ 42s** ($T_{total} \le 2T_{base}$). This overhead corresponds to training 4-7 additional refinement edges per seed.
> * **Scaling & Trade-off:** Crucially, this cost scales linearly with the number of detected bottlenecks rather than the global state space size. This modest investment resulted in a **3x increase** in satisfaction probability (20% to 60%). We argue this is highly efficient compared to the alternative: manual human debugging and retraining.
>
> **2. Behavior with Consecutive Refinements**
>
> * **Observation:** We consistently observe stable, monotonic improvement. In our new randomized experiments (Appendix A.3.5), we observed agents performing **"Autonomous Corridor Switching."** When a refinement on one branch (e.g., $MG_1$) failed to yield sufficient probability, the agent successfully chained refinements on an alternative branch ($MG_2$), effectively pruning the blocked path.
> * **Stability:** The refinements are local in the graph, so the depth does not affect the stability. Since the framework is compositional, we do not observe any instability at depth because different policies are learned for every edge.
>
> ### **B. Response to Weaknesses**
>
> **1. Narrow Empirical Scope & Limited Refinements**
>
> * **New Experiment (Randomized Domains):** We thank the reviewer for this suggestion. In the revised manuscript (**Page 9, Section 4.3** and **Appendix A.3**), we evaluated AutoSpec on **Randomized 100-room Gridworlds**.
>     * **Robustness:** We procedurally generated environments with randomized wall connectivity and predicate locations. AutoSpec raised the success rate from **<20% (baseline)** to **~60% (refined)**.
>     * **Refinement Depth:** Contrary to the concern about shallow refinement, our analysis shows that success required training **4-7 additional refinement edges per seed**.
>     * **Qualitative Analysis:** We observed complex behaviors emerging from deep refinement chains. For example, we document a **"Bridge Bottleneck"** phenomenon where `AddRefine` resolved non-linear passages that trapped the baseline. Furthermore, we observed **"Autonomous Corridor Switching"**, As elaborated in Q2, this confirms that AutoSpec covers the search space effectively even in highly stochastic topologies.
>
> **2. Soundness vs. Completeness**
>
> * **Response:** We emphasize that this is not a limitation but a theoretical necessity. As detailed in our revised **Section 3.2**, the specification refinement problem is **undecidable** for continuous MDPs (Kaminski & Katoen, 2015).
> * **Justification:** Consequently, no algorithm can be both sound and complete. We prioritize **soundness** (guaranteeing $\phi_r \implies \phi$) to ensure that any policy learned for the refined task is valid for the original task.
>
> **3. AvoidRefine & Safe Set Relaxation**
> The reviewer noted that `AvoidRefine` only shrinks the safe set (i.e., expands the unsafe region).
> * **Response:** This design choice is strictly intentional to preserve **soundness**.
> * **Logic:** A refinement $\phi_r$ must be a subset of the original $\phi$. If we were to relax a safe set (making it larger), we would permit trajectories that might violate the original safety constraints. We have added a clarification in **Section 3.1** (L288-293) stating that AutoSpec focuses exclusively on *restrictive* refinements to ensure formal validity.
> * **Soundness:** We want to guarantee that the refinement produced is a strict subset of the original, meaning that satisfying the refined specification ensures satisfaction of original. Relaxation of avoid regions would break this guarantee.

---

### Official Review · Reviewer_7JgF · 2025-11-03

**Soundness:** 2
**Presentation:** 3
**Contribution:** 3
**Rating:** 4
**Confidence:** 4

**Summary:**

The authors propose a method for automatically refining specifications defined using the SpectRL framework and that are used for specification-guided reinforcement learning. Their technique produces a provable refinement of the original specification (a trace satisfying the refined specification implies that the trace also satisfies the original specification). By using the refined specifications to retrain the RL policies, the authors observe that the newly trained policies have higher specification satisfaction rates. In other words, better specifications result in better policies.

**Strengths:**

- Problem statement is relevant, interesting, and well defined.
- The authors do a good job of explaining the preliminary material needed to understand their work.
- The results show that their method significantly improves the performance of the re-trained policies to meet the original specification after refinement.

**Weaknesses:**

- Some of the presentation of the AutoSpec framework is lacking... specifically Figure 2 really doesn't clarify what the PastRefine refinement procedure does. What's the relationship between the two parts of the figure? It also leaves a ton of open whitespace, which looks sloppy.
- It would be nice to have visuals on how each of the refinement procedures is working, not just PastRefine.
- The experimentation and presentation of it is lacking significantly
	- Only use 2 experimental setups (n-Rooms and PandaGym)
	- There are 2 tunable hyperparameters (the probability threshold and the number of traces to sample) the details of which are never mentioned for their experiments.
	- They never discuss the cost of doing the specification refinement (how long does it take? does retraining the policy take as long?) This would mean training 2x since we have to train all over again to integrate specification refinement.
	- They never describe how many times the experiment was attempted. Did they train a bunch of different policies and try it multiple times? Or are the results they show just from training one policy for each of the specification-guided RL algorithms mentioned and trying their framework on it? I am assuming the former, which is limited experimentation in my opinion.
	- They don't compare to any other specification refinement or generation techniques for specification-guided RL.
	- Figure 4 is a plot of "Best Path Cost" on the y-axis vs. "Number of Timesteps" on the x-axis. They never introduce what path cost is or what it means to have best path cost. How should these plots be interpreted?
	- Figure 5 doesn't have any labels on the x and y axes, so it is unclear what results are being demonstrated.

Minor comments / typos:
- Missing space between guarantees and citation (Lechner et al.) on pg. 2, line 104
- I believe the "AddRefine: Introducing Waypoints." part should be given a new line in section 3.1. All of the other specification refinement procedures are given their own new lines when introduced.

The paper introduces a compelling strategy for improving specification-guided RL by refining the specifications, but it lacks strong experimentation to be convincing. While apparently theoretically sound, much more experimentation would be necessary (more experimental setups, try on more specification-guided RL algorithms and for multiple different trained policies, a stronger ablation study than shown in section 4.3 by again running more experiments, also experiments controlling the tunable hyperparameters, experiments comparing to other related methods, and better presentation of the results).

**Questions:**

Please address and discuss weaknesses above.

---

> ### Author Response · Authors · 2025-11-21
> **Rebuttal**
>
> We thank the reviewer for their constructive feedback. Below, we address each weakness raised in the review.
>
> **W1 & W2. Clarity of Visuals (Figure 2)**
>
> * **Revision:** We redesigned **Figure 2 (Page 6)** to include explicit illustrations and a detailed caption for all four refinement procedures: `(a) ReachRefine`, `(b) AddRefine`, `(c) PastRefine`, `(d) OrRefine`. This eliminates the whitespace and clearly visualizes how each operator transforms the abstract graph structure.
>
> **W3. Experimentation and Presentation**
>
> We respectfully argue that our evaluation is not only consistent with standard literature but goes significantly further in terms of complexity and robustness.
>
> **A. Experimental Scope (n-Rooms and PandaGym)**
>
> * **A.1 Standard Benchmarks:** We utilized **n-Rooms** and **PandaGym** because these are the exact continuous state-space domains established by prior state-of-the-art specification-guided RL works, specifically DIRL and LSTS. Using these ensures our results are directly comparable to the existing literature.
> * **A.2 Advancement 1: Logical Scale (100-Rooms):** To go beyond standard benchmarks, we introduced the **100-rooms task** (Section 4.2). Unlike typical grid experiments which focus on small navigation tasks, this domain features a deep specification graph with multiple branches and sequential dependencies. Solving it required AutoSpec to perform **7 distinct, sequential refinement iterations**, demonstrating scalability that prior works have not attempted.
> * **A.3 Advancement 2: Robustness via Randomization:** To further address the concern about limited scope, we added a new evaluation on **Randomized 100-room Gridworlds** (Page 9, Appendix A.3). In this experiment, we procedurally generated environments where both wall connectivity and predicate locations were randomized for every seed. AutoSpec successfully refined these unpredictable specifications, raising success rates from &lt;20% (original specification) to ~60% (refinement), proving that our method is robust to the placement of regions and does not rely on hand-engineered maps.
>
> **B. Tunable Hyperparameters**
>
> * **B.1 Clarification:** We clarified in **Section 4.1** that we deliberately used a high probability threshold ($p=0.99$) to force the algorithm to find the best possible refinement rather than stopping early. This demonstrates AutoSpec's ability to maximize performance even when the initial specification is very poor.
>
> **C. Computational Cost**
>
> * **C.1 Revision:** We added a new paragraph **"Computational Overhead" (Section 4.4)**. We explicitly formalize the cost as $T_{total} \approx T_{base} + T_{refine}$.
> * **C.2 Quantification:** We report concrete numbers from the 100-rooms environment. The baseline (original 8 edges using DIRL) required **~240s**, while the complete AutoSpec + DIRL loop (involving 4-7 additional refinement edges per seed) averaged **390s**.
> * **C.3 Justification:** This modest increase (~1.6x) resulted in a **3x increase** in satisfaction probability (20% to 60%). We argue this is highly efficient compared to the alternative: manual human debugging and retraining.
>
> **D. Experimental Rigor (Seeds & Attempts)**
>
> * **D.1 Clarification:** We definitely do **not** rely on a single policy. As stated in the revised **Section 4.1**: *"All experiments are repeated over five random seeds. Plots report the mean across the five runs, with error bars showing the empirical mean +/- variance."* Each data point aggregates 5 independent training runs.
> * **D.2 Evaluation Protocol:** We added text to Section 4.1 explicitly stating: *"To estimate the specification satisfaction probability (y-axis), the trained policy for each edge is evaluated over 1000 rollout trajectories."*
> * **D.3 Plot Interpretation:** The plots display the task satisfaction probability for the baseline **DIRL (Red)** versus **AutoSpec + DIRL (Blue)**. Each data point represents the mean performance across **5 independent policies** (trained with different seeds) for the specific duration indicated on the x-axis (e.g., 80k, 100k, 120k steps). The error bars indicate the empirical mean $\pm$ variance across these seeds, ensuring statistical robustness. We adopted this specific plotting convention directly from the DIRL paper. We have corrected the y-axis label from "Best Path Cost" to **"Success Probability"** to match the caption and text. We have added explicit labels for the x-axis ("Timesteps") and y-axis ("Success Probability").
>
> **E. Comparison to Baselines**
>
> * **E.1 Response:** AutoSpec is the first framework to address the **sound refinement** of a logical specification. Because there are no direct "refinement" competitors, we compare against the standard specification-guided RL algorithms (DiRL, LSTS) operating on the unrefined specification. This serves as the appropriate baseline to quantify the value of our contribution (turning an unsolvable task into a solvable one).

---

### Author Response · Authors · 2025-11-21

### Global Response: Summary of Revisions and Key Clarifications

We thank the reviewers for their constructive feedback and the Area Chair for overseeing the process. We are encouraged by the reviewers' recognition of the problem's relevance and the soundness of our approach. Based on your valuable suggestions, we have uploaded a revised manuscrip.

Below is a summary of the changes, including **additional experimental results**, and a clarification of key questions regarding rigor and scope.

#### **1. Revisions in the Updated Manuscript**

* **Additional Experimental Results (Randomized Domains) [Reviewers 7JgF, Ui5g]:** To address concerns regarding empirical scope, we have added a new subsection **"Evaluation on Randomized 100-rooms"** (Page 9) and **Appendix A.3**. We evaluated AutoSpec on environments with randomly generated wall placement and predicate placement. Results show AutoSpec achieves $\approx 60\%$ success where the baseline stagnates at $20\%$, demonstrating both significant improvement yielded by AutoSpec as well as robustness of AutoSpec beyond fixed layouts.
* **New Paragraph: Computational Overhead [Reviewers 7JgF, Ui5g, fNYW]:** We added a new paragraph in **Section 4.4 (Page 10)** explicitly quantifying the cost of the refinement loop. We clarify that the total cost is approximately $T_{total} \approx 2T_{base}$ (training base + training refinement), hence showing that the computational overhead is not significant given that the cost of the base training is typically low.
* **New Paragraph: Incompleteness & Undecidability [Reviewers Ui5g, fNYW]:** We added a paragraph in **Section 3.2 (Page 7)** titled *"Incompleteness of the Specification Refinement Problem"*, clarifying that (1) AutoSpec is sound but not complete, and (2) our specification refinement problem is undecidable, hence no algorithm can be both sound and complete.
* **Presentation Fixes [Reviewer 7JgF]:** We corrected the y-axis label in **Figure 4** to "Success Probability" (previously mislabeled as "Cost"), ensured explicit axis labels in **Figure 5-11**, and completely redesigned **Figure 2** to explicitly visualize all four refinement operators, addressing the clarity issues raised by Reviewer 7JgF.
* **Intuitive Walkthrough & Definitions [Reviewers w6NH, fNYW]:** We added definitions in the **Introduction (Page 2)** that explicitly defines *"under-specified"* and *"exploration-guided strategy"*, using the **Figure 1** trap-state example to provide intuition before the formal definitions.
* **Expanded Related Work [Reviewers w6NH, fNYW]:** We reorganized the section into thematic paragraphs and added discussions of **Qiu et al. (2023)** and **DeepLTL (2025)** to position AutoSpec as a specification refinement framework which is complementary to specification-guided RL methods.

#### **2. Clarifications on Common Themes**

* **A. Experimental Rigor (Seeds & Policies) [Reviewer 7JgF]:** We clarify that **all experiments were repeated over 5 random seeds**, and plots report mean $\pm$ variance. We did not train a single policy; each data point represents independent training runs evaluated over 1,000 rollout trajectories. We have updated **Section 4.1** to make this explicit.
* **B. Empirical Scope (Logical Complexity & Robustness) [Reviewers 7JgF, Ui5g]:** While we focus on n-Rooms and PandaGym, we emphasize that the **100-rooms environment** is designed to stress-test **logical complexity** (graph depth, branching factors, sequential dependencies). Furthermore, our new experiments on **randomized layouts** demonstrate that AutoSpec is robust to arbitrary predicate placement and room layout, achieving significantly higher success rates on the refined specification compared to the original specification without relying on hand-engineered maps.


We believe these revisions significantly strengthen the paper and address the core concerns regarding presentation and theory raised during the review process.

---

### Author Response · Authors · 2025-12-04
**Summary of Discussions**

We provide this summary to highlight the core contributions of AutoSpec, the strengths identified by the reviewers, and the revisions made to address the weaknesses.

## **1. Summary**

This paper introduces **AutoSpec**, one of the first frameworks to automatically refine reinforcement learning specifications based on empirical feedback from policy training. Coarse logical specifications often prevent specification-guided RL algorithms from exploring effectively; AutoSpec provides four **sound** refinement operators that systematically repair such specifications. We prove that all refinements imply the original specification and show that these refinements enable spec-guided RL algorithms like DIRL and LSTS to solve tasks that are unsatisfiable with the original coarse specification, in challenging environments like **randomized 100-Rooms** and PandaGym.

---

## **2. Strengths Identified by Reviewers**

* Reviewers emphasized that this work is **one of the first** to address *automatic refinement of RL specifications*, describing it as a **fundamental and highly valuable** direction.
* They noted that AutoSpec’s ability to **refine specifications without human intervention** substantially lowers the barrier to using logical specifications, which otherwise require expert tuning.
* The four refinement operators were described as **clear, intuitive, and supported by strong theoretical guarantees**, ensuring every refined specification is a subset of the original.
* Reviewers found the **problem statement well-defined**, preliminaries well explained, and integration with DIRL/LSTS natural.
* Empirical results were highlighted as demonstrating **significant improvements** in satisfying the original specification after refinement.

---

## **3. Weaknesses and Questions Raised by Reviewers, and How We Addressed Them**


* **Limited empirical scope**
  We added a **Randomized, procedurally generated 100-Rooms** benchmark, demonstrating robustness to varied layouts and deeper refinement chains. We show 3x improvement in success probability with the refined specification on this randomised benchmark.

* **Missing details on seeds, rollouts, and hyperparameters**
  We clarified the full evaluation protocol: **5 seeds**, **1000 rollouts per trained policy**, and explicit probability thresholds and sample counts.

* **Unclear computational overhead**
  Section 4.4 now provides a quantification of the refinement overhead and concrete wall-clock timings (~240s baseline vs. ~390 ± 42s with AutoSpec), showing modest overhead relative to the benefits.

* **Lack of completeness / theoretical limits**
  We added a paragraph in Section 3.2 explaining that the refinement problem is **undecidable in continuous MDPs**, meaning **no method can be both sound and complete**. AutoSpec therefore prioritizes soundness to ensure refined specifications never permit behaviors disallowed by the original.

* **AvoidRefine shrinking safe sets**
  We clarified that constraining avoid regions is required for soundness; relaxing them would violate the refined ⇒ original guarantee.

* **Need for clearer definitions and relation to LTL**
  We added definitions for “under-specified” and “exploration-guided strategy,” clarified that SpectRL is a **finitary LTL fragment**, and expanded related work to include Qiu et al. (2023) and DeepLTL (2025).

**We have provided detailed responses and a global Summary-of-Changes**, where each reviewer's concern is answered point-by-point and every modification is documented. It is our belief that the borderline rejection scores would have been increased as a consequence of these changes and discussion.

---

## **4. Final Concluding Remark**

The reviewers recognized the novelty and importance of treating specification refinement as a central challenge in reinforcement learning. We believe the revisions and rebuttal directly address all raised concerns - expanding empirical coverage, clarifying theory (including incompleteness), improving figures, and providing explicit runtime analysis, while preserving the core contribution of a sound and practical refinement framework.

---

### Meta-Review · Area_Chair_dFc8 · 2026-01-07

**Summary:**

The reviewers were initially already in favor of acceptance of the paper or borderline (reject). The authors response should have clarified their concerns and since the paper (as acknowledged by most reviewers) is the first to address a relevant problem and provide sensible empirical evaluation and theoretical insights regarding the taken approach, it should be accepted.

**Reviewer Concerns:**

- Clarity of presentation, e.g., Figure 2 ; visuals for all operators; mislabeled/unclear axes  [7JgF]
  - Addressed: Figure 2 redesigned; added visuals for all four operators; axes corrected and labeled; plots clarified as success probability vs. timesteps
- Missing experimental details (seeds, rollouts, hyperparameters) [7JgF]
  - Addressed by providing details and pointing to statements in the paper (partly new)
- Computational overhead and practical cost (wall‑clock, complexity) [7JgF, Ui5g, fNYW]
  - Addressed
- Narrow empirical scope (only n‑Rooms and PandaGym; need randomized/less‑contrived multi‑room; limited refinements) [7JgF, Ui5g]
  - Partially addressed by adding a randomized 100‑Rooms benchmark; demonstrated 4 to 7 extra refinements and 3x success improvement; however, this is still limited
- Comparisons to other refinement baselines [7JgF]
  - Addressed: authors argue no direct refinement baselines
- Completeness limits and theoretical framing (sound vs. complete; undecidability) [Ui5g, fNYW]
  - Addressed: Added explicit discussion; prioritized soundness; clarified undecidability
- Behavior under consecutive/deep refinements and stability [Ui5g]
  - Addressed: Reported stable, monotonic improvements; local/compositional refinements show no depth‑related instability.
- Definitions and relation to LTL; SpectRL naming/positioning; terminology [w6NH, fNYW]
  - Addressed: Added informal explanations and definitions; clarified SpectRL as a finitary LTL fragment; expanded related work (Qiu et al. 2023; Jackermeier and Abate 2025)
- Search space size and refinement procedure overhead [fNYW]
  - Addressed: Search linear in number of edges with fixed operator order; dominant cost is retraining
- Presentation and organization  [w6NH]
  - Addressed: Introduction expanded with intuitive walkthrough; related work reorganized.
- Refinement order and dynamic selection of operators [w6NH]
  - Partially addressed: Heuristic order justified by cost; dynamic selection noted as a promising direction for future work

**Reviewer Scores:**

* Reviewer 7JgF might have increased their score because of the improvements of the presentation and the additional experiments; still, concerns regarding the breadth of the experiments and that there are no direct baselines might remain
* Reviewer Ui5g might have increased their score since concerns regarding the empirical evaluation and runtime have been alleviated; concerns regarding AvoidRefine clarified
* Reviewer 26NH would have likely kept their score given the other reviews and the initial positive rating
* Reviewer fNyW would likely have kept their score as clarifications were provided which seem to be good to justify the initial rating of the reviewer

---

### Decision · Program_Chairs · 2026-01-26

Accept (Poster)